# Propagating Rifts: The Roles of Crustal Damage and Ascending Mantle Fluids

Folarin Kolawole[1,2], Rasheed Ajala[1]

[1]Lamont-Doherty Earth Observatory of Columbia University, Palisades, New York, 10964, United States
[2]Department of Earth and Environmental Sciences, Columbia University, New York, 10027, United States

*Correspondence to*: Folarin Kolawole (fola@ldeo.columbia.edu)

**Abstract**

We investigate the upper-crustal structure of the Rukwa-Tanganyika Rift Zone, East Africa, where the Tanganyika Rift is interacting with the Rukwa and Mweru-Wantipa rift tips, manifested by prominent fault scarps and seismicity across the rift interaction zones. We invert earthquake P and S travel times to produce three-dimensional upper-crustal velocity models for the region and perform seismicity cluster analysis to understand strain accommodation at rift interaction zones and the propagating rift tips. The resulting models reveal the occurrence of anomalously high Vp/Vs ratios in the upper-crust beneath the Rukwa and Mweru-Wantipa rift tips — regions with basement exposures and sparse rift sedimentation. We detect distinct earthquake families within the deeper clusters which exhibit an upward linear temporal evolution pattern that suggests triggering by upward fluid migration and creep failure. A spatial transition from proximal tip zones dominated by thinned crust and through-going crustal and upper-mantle seismicity to distal tip zones with thick crust and dominantly upper-crustal seismicity indicate an along-axis variation in the controls on rift tip deformation. Overall, the collocation of basement faulting, crustal and upper mantle seismicity, and upper-crustal high Vp/Vs ratios suggest a mechanically weakened crust at the rift tips, likely accommodated by brittle damage from crustal bending strain and thermomechanical alteration by ascending fluids (mantle-sourced volatiles, and hydrothermal fluids). These findings provide new insights into the physics of continental rift tip propagation, linkage, and coalescence — a necessary ingredient for initiating a continental break-up axes.

## 1 Introduction

The mechanism of segmentation and lateral propagation and linkage of continental rifts, first introduced by Bosworth (1985), has received significant attention from the scientific community as they influence the structure and temporal progression of the evolving break-up axis (e.g., Ebinger et al., 1989, 1999; Nelson et al., 1992; Acocella, 1999; Aanyu and Koehn, 2011; Allken et al., 2012; Corti, 2004; Zwaan et al., 2016; Neuharth et al., 2021; Kolawole et al., 2021a; Brune et al., 2023). Previous studies have established that continental rift systems grow by initial nucleation of isolated segments that propagate laterally, interact, link up, and coalesce to form longer composite rift basins with a continuous rift floor. Prior to linkage, the propagating rift segments are separated by an 'unrifted' crustal block, and the lateral propagation of the rift deformation into the intervening block is essential to advance the rift system towards break-up (e.g., Nelson et al., 1992; Kolawole et al., 2021a; Brune et al., 2023).

In regions of active tectonic extension, inelastic deformation manifests by tectonic and magmatic deformation of the crystalline crust and its overlying sedimentary sequences in the rift basins (e.g., Brune et al., 2023; Pérez-Gussinyé et al., 2023). However, in magma-poor (i.e., non-volcanic) active rift settings, tectonic deformation in continental rifts is commonly accommodated by widespread brittle deformation of the crust through faulting and fracturing and accompanied by earthquakes (e.g., Muirhead et al., 2019; Kolawole et al., 2017, 2018; Gaherty et al., 2019; Zheng et al., 2020; Stevens et al., 2021). Nevertheless, little is known of how this deformation is transferred onto the propagating rift tips, and long-standing questions remain on how the earth's crystalline crust accommodates and localizes tectonic strain during continental rift propagation.

In this study, we use recently acquired seismic data to explore the upper crustal structure of the Rukwa-Tanganyika Rift Zone (Fig. 1a), an active non-volcanic rift zone along the East African Rift System, where previous studies have suggested a thick, strong lithosphere (Craig et al., 2011; Foster and Jackson, 1998; Yang and Chen, 2010; Hodgson et al., 2017; Lavayssière et al., 2019) and ongoing unilateral propagation of the Rukwa Rift tip (Kolawole et al., 2021a). A previous study (Hodgson et al., 2017) utilized the receiver function technique to map the spatial distribution of crustal-averaged Vp/Vs ratios but lacked constraints on the shallowest structure. Our results provide insight into the fundamental mechanism of strain distribution and localization along actively propagating rift segments. Ultimately, the approach may advance our understanding of how incipient divergent plate boundaries mature within active continental environments.

## 2 The Rukwa-Tanganyika Rift Zone

### 2.1 Pre-Rift Crystalline Basement

The crystalline crust of the Rukwa-Tanganyika Rift Zone (Fig. 1a) is mainly composed of metamorphic and igneous rocks of the Paleoproterozoic (1.85–1.95 Ga) Ubendian mobile belt (Fig. 1b), flanked by Archean crystalline rocks of the Bangweulu and Tanzania cratons and their overlying Neoproterozoic sedimentary sequences to the southwest and northeast respectively (Fig. 1b). The Ubendian Belt consists of several amalgamated NW-trending terranes defining the orogenic belt that accommodated the Paleoproterozoic collision events (2.025–2.1 Ga) between the Tanzania Craton and the Bangweulu Block. The terranes, comprising Ufipa, Katuma, Wakole, Lupa, Mbozi, Ubende, and Upangwa (Fig. 1b; Daly, 1988; Lenoir et al., 1994), are now exhumed due to long-term erosion and are bounded by steeply-dipping, ductile, amphibolite facies, strike-slip shear zones (Fig. 1b; Daly, 1988; Lenoir et al., 1994; Theunissen et al., 1996; Kolawole et al., 2018, 2021b; Lemna et al., 2019; Heilman et al., 2019; Ganbat et al., 2021). Their associated ductile fabrics are suggested to have influenced the development of post-Precambrian rift basins in the region (Wheeler and Karson, 1994; Theunissen et al., 1996; Klerkx et al., 1998; Boven et al., 1999; Heilman et al., 2019; Lemna et al., 2019; Kolawole et al., 2018, 2021a,b).

### 2.2 Phanerozoic Rifting History

The Rukwa-Tanganyika Rift Zone is defined by a system of NNW-to-NW-trending overlapping rift segments, consisting of the Tanganyika Rift, the Rukwa Rift to its southeast, and the ENE-trending Mweru-Wantipa Rift located just southwest of Tanganyika's southernmost sub-basin (Figs. 1a-b). The rift zone records multiple phases of Phanerozoic tectonic extension, with the first phase occurring in the Late Permian to Triassic, the second phase beginning in the Late Jurassic but peaking in the Cretaceous, and the third phase initiating in the Late Oligocene and presently persisting (e.g., Delvaux, 1989, Roberts et al., 2012). Although studies show that all the rift segments are currently active (e.g., Daly et al., 2020; Hodgson et al., Lavayssiere et al., 2019; Heilman et al., 2019; Kolawole et al., 2021a), not all the basins record the three phases of Phanerozoic rifting (Delvaux, 1989; Morley et al., 1992, 1999; Muirhead et al., 2019; Shaban et al., 2023). Within the rift zone, the Rukwa Rift is the only basin with basement-penetrating borehole logs to constrain seismic reflection interpretation, producing detailed mapping of the lateral extents of the Mesozoic and Cenozoic syn-rift sequences (Morley et al., 1992) and relationships with rift faulting patterns (Morley et al., 1992, 1999; Heilman et al., 2019; Kolawole et al., 2021b). The distribution of the syn-rift deposits and faulting patterns show that the Rukwa Rift progressively elongated northwestwards and southeastwards over its polyphase extensional tectonic history (Morley et al., 1999; Heilman et al., 2019; Kolawole et al., 2021b).

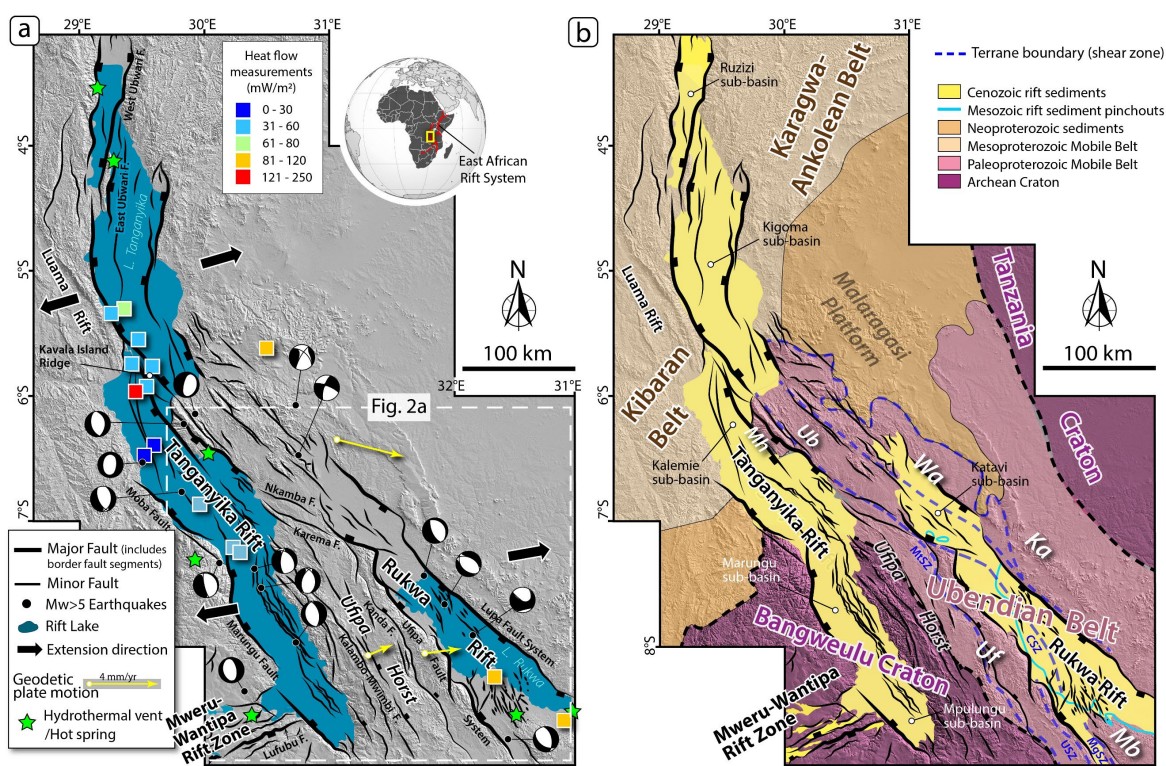

**Figure 1**. (**a**) Tectonic map of the Rukwa-Tanganyika Rift Zone showing the rift faults (Morley et al., 1999; Muirhead et al., 2019; Kolawole et al., 2021a). Focal mechanisms and epicenters of Mw >5 earthquakes from National Earthquake Information Center (NEIC) catalog (1976–2018) obtained through the United States Geological Survey website (https://earthquake.usgs.gov/earthquakes/search/). Geodetic plate motion vectors are from Stamps et al. (2008). Regional extension directions are from Delvaux and Barth (2010) for the northern Tanganyika

Rift and Lavayssière et al. (2019) for the southern Tanganyika and Rukwa rift basins. Heat flow measurements and their locations are from
Jones (2020). Sites of hot springs/hydrothermal vents are from Tiercelin et al. (1993), Lavayssière et al. (2019), Jones (2020), and Mulaya
et al. (2022). (**b**) Geological map of the region, showing the cratons, mobile belts, terranes of the Ubendian Belt and shear zones, and
Cenozoic syn-rift sediments (modified after Daly, 1988; Hanson, 2003; Delvaux et al., 2012; Kolawole et al., 2021a,b; Ganbat et al., 2021).
Ubendian Belt Terranes: Ka - Katuma, Mb - Mbozi, Mh - Mahale, Ub - Ubende, Uf - Ufipa, Wa - Wakole. Exhumed Precambrian shear
zones (Delvaux et al., 2012; Heilman et al., 2019): CSZ, Chisi Shear Zone; MgSZ, Mughese Shear Zone; MtSZ: Mtose Shear Zone; USZ:
Ufipa Shear Zone.
The Cretaceous rifting event included reactivated faulting, tectonic subsidence, and sedimentation in the Rukwa Rift and
Luama Rift (e.g., Veatch, 1935; Delvaux, 1991; Roberts et al., 2012). Cenozoic rifting initiated the development of rift basins
as segments of the East African Rift System, featuring the reactivation of the Rukwa Rift and the development of the
Tanganyika and the Mweru-Wantipa rift segments (e.g., Morley et al., 1999; Delvaux et al., 2001; Chorowicz, 2005; Daly et
al., 2020). Crustal thickness across the rift zones range 31.6 – 42 km (Hodgson et al., 2017; Njinju et al., 2019) and lithosphere
thickness 130 – 170 km (Njinju et al., 2019). The contemporary regional minimum compressive stress orientation is 074° in
the northern Tanganyika Rift (Delvaux and Barth, 2010) and 080° in the southern Tanganyika and Rukwa rifts (Lavayssière
et al., 2019) (Fig. 1a). Although contemporary regional stress in the Mweru-Wantipa Rift is unknown, the Mweru Rift, which
is its southwestern continuation, is shown to have a regional minimum compressive stress orientation of 118° (Delavaux &
Barth, 2010).

## 2.3 Rift Faulting and Seismicity Patterns

The Tanganyika Rift basin is bounded by a system of large border faults that alternate polarity along-trend of the basin (Versfelt
and Rosendahl, 1989) and include the Marungu Fault, the Kavala Island Ridge Faults, the West and East Ubwari Faults, and
the Moba Fault (Fig. 1a), whereas the large graben of the Rukwa Rift basin is bounded by laterally continuous border fault
systems of the Lupa Fault to the northeast and Ufipa Fault to the southwest (Heilman et al., 2019). The Ufipa Horst represents
the intervening basement block between the southern Tanganyika Rift and the Rukwa Rift and is accommodating active
deformation as evidenced by the ca. 100-km long scarps of the Kanda and Kalambo-Mwimbi Faults (Fig. 1a; Delvaux et al.,
2012; Kolawole et al., 2021). Moreover, two prominent fault scarps extend WNW from the Rukwa Rift tip across a basement
region to the eastern margins of the central Tanganyika Rift (Nkamba and Karema Faults; Fig. 1a). To the southwest, the
deformation zone of the Mweru-Wantipa Rift hosts a ca. 50-km-wide parallel fault cluster that defines its southeastern margin
within which the Lufuba Fault appears to have the greatest escarpment height (Fig. 1a).
The entire Rukwa-Tanganyika Rift Zone records widespread seismicity (Figs. 2a, c–d) that extends beyond 42 km depth,
indicating that the seismogenic layer of the rift includes the uppermost mantle (Fig. 2c–e; Lavayssière et al., 2019). The events
define clusters with focal mechanism solutions that suggest steep, deep-rooting, large normal faults (Lavayssière et al., 2019),
and highlight localized active crustal deformation zones beneath Tanganyika Rift, Rukwa Rift, the Ufipa Horst, and the
Mweru-Wantipa Rift (Fig. 1a). Across the rift zone, the earthquakes commonly continue down into the lower crust; however,
beneath the northwestern tip of the Rukwa Rift (Katavi sub-basin; Figs. 2a, 2d) the earthquakes occur in both the upper crust
and upper mantle (Lavayssière et al., 2019). More interestingly, the axis of the Rukwa Rift has sparse seismicity. Seismicity
clusters at the Rukwa Rift tip extend beyond the margins of the basin sediments, continuing outboard into the regions of the
exposed pre-rift basement (Figs. 2a and 2d). In the southern Tanganyika Rift, earthquakes mostly cluster within and along the
rift axis (Figs. 2a and 2c). Previous seismic receiver function and crustal anisotropy studies show evidence indicating the
presence of partial melt/volatiles in the lower crust (Hodgson etal., 2017; Ajala et al., 2024), and demonstrate how lower crustal
fluids promote strain localization (Ajala et al., 2024). Heat flow measurements in the rift zone show thermal anomalies in the
central Tanganyika Rift (<30 to 250 mW/m2), the south-central region of the Rukwa Rift (81 – 120 mW/m2), and within the
basement region ahead of the northwestern tip of the Rukwa Rift (81 – 120 mW/m²) (Fig. 1a; Jones, 2020). The thermal
anomaly north of the Rukwa Rift tip occurs near NW-trending fault splays and Mw>5 earthquake epicenters within the
basement region. Furthermore, hydrothermal vent and hot spring locations coincide with the border fault zones of the
Tanganyika Rift and the south-central part of the Rukwa Rift (Fig. 1a; Tiercelin et al., 1993; Lavayssière et al., 2019; Jones,

154 2020).

### 155 2.4 Active Deformation Across the Rift Interaction Zones

At a regional scale, the Rukwa and Tanganyika rift basins are separated by an elevated region of pre-rift basement with
widespread exposures of Precambrian metamorphic rocks (Figs. 1a-b; Kolawole et al., 2021a). This elevated region of rift
overlap includes the Ufipa Horst to its south, and the region between the northern tip of the Rukwa and the eastern flank of the
central Tanganyika Rift to its north. In a geodynamic context, the geometry of the overlap region defines an overlapping
parallel-to-oblique 'rift interaction zone' (Kolawole et al., 2021a) and is characterized by historical seismicity and active faults
that deform the modern surface (Delvaux et al., 2001; Lavayssière et al., 2019; Kolawole et al., 2021a). The faults include the
WNW-trending Karema and Nkamba faults, which splay westwards from the Rukwa Rift tip (Fig. 1a; Fernandez-Alonso et
al., 2001; Kolawole et al., 2021a), and NW-trending faults that extend northwards towards the margin of the northern
Tanganyika Rift (Kolawole et al., 2021a). The longitudinal surface relief morphology of the southern Tanganyika Rift shows
a significantly steeper gradient than that of the Rukwa Rift tip ('rift tip' in Fig. 2c versus 2d). Overall, the current stage of
evolution of the rift interaction zone based on the relief profile, stream flow patterns, and drainage morphologies is inferred to
be partially breached (Kolawole et al., 2021a). To the southwest, the Mweru-Wantipa Rift extends eastward and appears to be
hard-linked with the border fault of the western flank of the southern tip of the Tanganyika Rift. The region between the two
rifts defines an overlapping orthogonal rift interaction zone, and the continuation of Lake Tanganyika into the Mweru-Wantipa
Basin and the apparent coalescence of the rift floors of the two basins suggest a breached rift interaction zone between them
(Kolawole et al., 2021a).

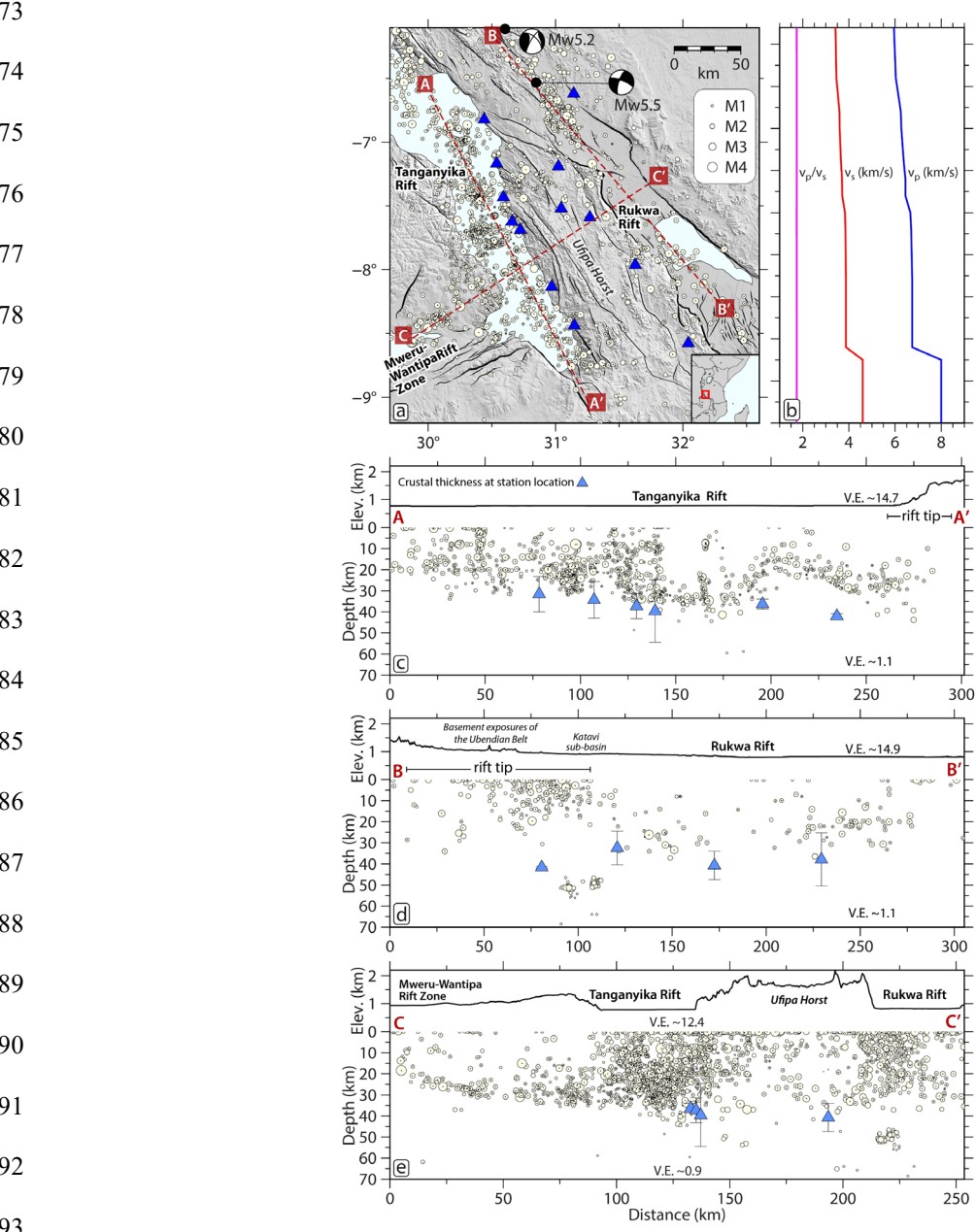

**Figure 2**. (**a**) Map of the southern Tanganyika and Rukwa rift zone showing the local seismicity (white circles; from TANGA14 array – network ZV, Lavayssière et al., 2019) scaled by magnitude. The red dots represent events used in the inversion. Blue triangles represent the locations of the TANGA14 broadband seismometers; Black lines are faults; the thicker black lines highlight border faults; Red dashed lines are locations of seismicity profiles in c-e. Inset map shows the relative location in East Africa. (**b**) Starting model used in the seismic tomographic inversion (from Lavayssière et al., 2019). (**c - e**) Elevation and depth profiles showing projected seismicity (from Lavayssière et al., 2019) and estimated Moho depths (from Hodgson et al., 2017) along and across the rifts. Cross-sectional profiles A-A' and B-B' only show earthquakes within 25 km on both sides of the cross-section traces, and all Moho depth plots are projected from stations located within 50 km of the profiles.


## 3 Data and Methods

### 3.1 Seismic Data

We focus on waveform data recorded by the TANGA14 array, comprising 13 broadband seismographs deployed along the
Ufipa Plateau for 15 months from June 2014 through September 2015 (Fig. 2a; Hodgson et al., 2017). Using the origin times
from the local earthquake catalog developed by Lavayssière et al. (2019) comprising 2213 events (Fig. 2a), we download the
associated waveforms using the facilities of the EarthScope Consortium. The waveforms were then filtered with a Butterworth
filter to accentuate the earthquake signal in the frequency band used in Lavayssière et al. (2019): 2 – 15 Hz. Arrival times for
both P and S waves were manually picked on filtered seismograms resulting in 3187 P times from 1277 earthquakes (resp.
3121 S times from 1261 earthquakes). We only made the travel time picks when the phases were clear and impulsive. We do
not record uncertainty in arrival times during picking, nor do we pick multiple times to estimate the data variance.

### 3.2 Crustal Imaging

#### 3.2.1 Backprojection Tomography

Using our manually picked P and S arrival times, we develop 3D P and S velocity models for the Tanganyika-Rukwa region
via nonlinear back-projection travel time inversion (Hole, 1992; Hole et al., 2000). For the study area, we use the 1-D P and S
velocity model developed by Lavayssière et al. (2019) as our initial velocity model. We parameterize the model space using a
fixed 5 km grid spacing with dimensions of 425 km x 435 km x 50 km. The bottom right corner of the model is 29.1651º E
and -9.6764º S and extends from 7 km above sea level to 43 km depth. Therefore, we use the actual station elevations without
needing static corrections. The travel time predictions in the model are calculated using a finite-difference solution for the
Eikonal equation (Vidale, 1990), which allows travel times to be computed for all grid points in the model. Ray paths are then
simultaneously traced for any number of source-receiver pairs using the gradient of the travel time field. Due to the reciprocity
in the travel time computation, we treat the receivers as sources, thus requiring only 13 forward computations in each iteration.
Following the forward calculation, we iteratively update the models, $k$, at each grid point, $j$, as follows:

$$u_{k+1}^j = u_k^j + \delta u_k^j, \tag{1}$$

where the slowness perturbations, $\delta u$, are calculated using simple back-projection as the average of the neighborhood ray
paths, i.e.,

$$\delta u_k^j = \frac{1}{N} \sum_{cells} \sum_{rays} \frac{\delta t_{ray}}{l_{ray}}, \tag{2}$$

with $\delta t$ and $l$ being the associated traveltime residual and raypath length for the associated ray. We further smooth the perturbations once they are determined for all grid points in the model using a 3D moving average filter to control the spatial resolution and stabilize the inversion. This procedure is like higher-order Tikhonov regularization in the least squares nonlinear inversion. We gradually reduce the size of the smoothing dimension after every five iterations to increase the spatial resolution of the model. The final smoothing size of our model from the 26th iteration is 5 x 5 x 3 grid points. Finally, the Vp/Vs ratio is obtained by dividing the P and S velocity models.

### 3.2.2 Model Reliability Assessment

To assess the model uncertainty, we employ a combination of ray coverage maps, classical checkerboard reconstruction tests, a custom synthetic model reconstruction test (targeted resolution test, e.g., Saeidi et al., 2024), and real data inversion using different starting models to determine areas of the model reliable enough for interpretation (Figs. 3 – 5 and S2 – S18). We generate the checkerboard models by adding 3D sinusoid functions to the initial velocity model (Fig. 2b) using similar magnitudes in the amplitudes of the real, inverted model (Fig. 3). The observed travel time dataset is computed in the checkerboard model and then inverted using the unperturbed starting model. We do not add noise to the synthetic datasets. We also test different sizes of the anomalies (Figs. S4 – S15). Based on the results from the artificial reconstructions, we define a polygon (e.g, Figs. S16 and 3) in the model space where the model parameters are reasonably resolved. Also, we developed and inverted a custom synthetic model (Figs. S16 and S17) based on the vital features we interpret in our final preferred model (Fig. 3) at the edge of the polygon where ray coverage is sparse or lacking (Fig. S2).

The synthetic model comprises three high Vp/Vs (~4 % increase) anomalies generated by perturbing the P (1 % increase) and S (3 % decrease) velocity model and extending from 2 km above zero to 13 km depth in the model space, with the following horizontal dimensions: 80 by 60 km at the north, 80 by 60 km at the southeast, and 65 by 85 km at the southwest. The inversion results show good recovery of the anomalies with some smearing outside our predefined polygon (Fig. S17). To further assess the reliability of these features in the real model (Fig. 3), we perform two other inversions of the real data using two different 3D initial velocity models from Celli et al. (2020) and van Herwaarden et al. (2023). A comparison of the results of all three starting models (Figs. S18 – S20) shows that the Vp/Vs anomalies are robust.

### 3.3 Seismicity Cluster Analysis

Visual inspection of the seismicity (Fig. 2) shows apparent spatial clusters. However, we must perform a spatiotemporal seismicity clustering analysis to determine which earthquakes are also close in time (Fig. S21). Although a complete statistical study of the earthquake catalog is beyond the scope of the current research, we perform a simple clustering analysis (Fig. S22) to highlight potential earthquake groups that could indicate fluid activity at the rift tips. First, we attempt catalog declustering to remove any aftershock sequences using the approach of Reasenberg (1985) as implemented in the CLUSTER2000 program. However, no significant aftershock sequences were found, with most aftershock clusters totaling 17 containing only two events

(Ajala & Kolawole, 2023). This is despite the earthquake frequency distribution (Fig. S23e) showing a decreasing amount of seismicity through time that would seemingly indicate the presence of aftershock sequences. The lack of correlation between the data availability periods when the seismic stations were operational (Fig. S23a) and the seismicity frequency (Fig. S23e) shows that there was indeed increased seismic activity during the earlier deployment times, particularly in August 2014. An enhanced earthquake catalog with a lower magnitude of completeness may be required in the region for declustering. Therefore, we decided to use the entire catalog as is in the clustering analysis. We analyze the earthquake catalog for clusters using the algorithm of Zaliapin et al. (2008) and Zaliapin & Ben-Zion (2013), as implemented by Goebel et al. (2019). For each event $j$ in the catalog, except for the earliest one, we find the parent event, which is an earlier event, defined using the smallest nearest-neighbor distance $\eta_{ij}$ computed to all the other events $i$ and defined as

$$\eta_{ij} = \begin{cases} t_{ij} r_{ij}{}^{d_f} 10^{-bm_i}, & t_{ij} > 0 \\ \infty, & t_{ij} \leq 0 \end{cases} \qquad (3)$$

where $t_{ij}$ is the time separation in years, $r_{ij}$ is the Haversine distance between the earthquake pairs epicenters, $d_f$ is the fractal dimension of the epicenters assumed to be 1.6 (Zaliapin et al., 2008), b is the Gutenberg-Richter b-value set to 1, and $m_i$ is the magnitude of the potential parent event $i$. To separate the nearest-neighbor distances into space $R_{ij}$ and time $T_{ij}$ components, we use the following relations,

$$T_{ij} = t_{ij} 10^{-qbm_i}, \qquad (4)$$

$$R_{ij} = r_{ij}{}^{d_f} 10^{-(1-q)m_i}, \qquad (5)$$

where we assume an interpolation factor $q$ of 0.5.

Finally, to split the catalog into background and cluster events, we estimate a separation threshold $\eta_0$ using the average of estimates of the 1st percentile of nearest-neighbor distances computed from 100 randomized catalogs with a similar range of space-time-magnitude parameters but with a Poissonian distribution representative of background seismicity (Fig. S22). At the estimated $\eta_0$, we see the probability distribution of the nearest-neighbor distances deviate from the Weibull probability distribution known to represent Poisson background seismicity (Fig. S22c; Zaliapin & Ben-Zion, 2013). Event pairs with nearest-neighbor distances less than $\eta_0$ that have similar parents are then recursively grouped to generate the clustered catalog (Figs. 4, S23b – d, and S24). For clusters at the Rukwa rift tip, we compute the normalized cross-correlation coefficients of the vertical component of the waveforms of events relative to the waveform of the parent event (Figs. 4g and h). We note that the lack of uncertainties in the earthquake catalog (Lavayssière et al., 2019) and relative earthquake locations may introduce spatiotemporal errors in the above analysis. In presenting our results, we use the time-magnitude plots as a guide to help distinguish between the mechanisms of the two swarms as either slow slip (creep) or fluid flow (Roland & McGuire, 2009).

## 4 Results

### 4.1 Crustal Seismic Velocity Models of the Tanganyika-Rukwa Rift Zone

We present the velocity models as perturbations (Fig. 3) relative to the starting models used in the inversion (Fig. 2b). The 5 km model grid spacing makes our selection of the 3 km depth maps (Figs. 3e, 3i, 4e, 4i, 5a, 5e, 5i) representative of the average uppermost crustal structure of the model in the region, as can be verified in the cross-sectional profiles of Figure 5. The overall distribution of upper crustal velocities generally reflects the near-surface geology, which serves as a primary constraint for assessing the quality of the models. Our results show that lower Vp and Vs are collocated with the sedimentary basins of the southern Tanganyika and Rukwa rifts. Relatively lower velocities continue along a narrow ESE-trending zone from the Tanganyika Rift to the northern end of the Rukwa Rift, following the Nkamba and Karema faults. The Ufipa Horst separating the Tanganyika and Rukwa rifts also shows localized zones of lower Vp, collocated with areas of prominent surface faulting (Fig. 3a). However, unlike the Vp distribution, the Ufipa Horst is better defined in the Vs model, demonstrated by the relatively higher values and structural continuity (Figs. 3e and h). Within the eastern section of the Mweru-Wantipa Rift and further east towards the southern Tanganyika Rift, we observe moderate Vp anomalies collocated with moderate-to-low Vs anomalies (Figs. 3a – b, e – f). Overall, the rift flanks and zones of widespread exposure of the pre-rift basement exhibit relatively higher Vp and Vs.

The Vp/Vs ratio map (Fig. 3i) and cross-sections (Figs. 3j – l) show zones of anomalously high values that are restricted to upper-crustal depths, the most prominent of which are A1: an anomaly at the northwestern end of the Rukwa Rift, an area dominated by basement exposures and distributed faulting, A2: a broad anomaly extending across the eastern end of the Mweru-Wantipa Rift through the transfer zone into the Tanganyika Rift, and A3: an anomaly in the southeastern interior of the Rukwa Rift, collocated with the Ufipa Fault and the intra-basement Chisi Shear Zone (Fig. 1b). These highest Vp/Vs anomalies commonly continue downward to 10 km or deeper (Figs. 3k – l) but our investigation focuses on the upper crust.

### 4.2 Spatiotemporal Clustering of Rift Tip Seismicity

Our cluster analysis yielded 115 clusters, but we only retained clusters with a minimum of 5 events, resulting in a filtered number of 18 clusters. The distribution of these clusters is shown as colored circles in map and cross-section views in Figures 4a–d and Figure S24 and as functions of latitude, longitude, and depth in Figures S23b – d. We identify clusters throughout the crust and in the upper mantle, with most of the clusters occurring along the intra-rift faults within the Tanganyika Rift. Some clusters are located at the tips of the Mweru-Wantipa and Rukwa rifts and on faults within the Ufipa Horst. Due to the focus of the current study on investigating rift tip processes, we only discuss the detected seismicity clusters at the Rukwa Rift tip, the three spatially clustered events occurring at 10 – 20 km depth at the Mweru-Wantipa Rift tip, and the absence of clean waveform records for these events preclude further analysis on these clusters.

There are two main clusters at the Rukwa rift tip (Figs. 4a – d), both occurring in the upper mantle between 40 – 70 km depths. The northern cluster comprises six events with local magnitudes between 0.67 and 1.35 that happened within a period of ~50 minutes on July 9, 2014 (Fig. 4e). In contrast, the southern cluster has 12 events with magnitudes between 1.2 and 2.8 that occurred within a period of ~19 days in June 2015 (Fig. 4f). We note the high waveform similarity of the events as recorded at nearby stations (Figs. 4g and h). In general, both clusters define a generally linear trend with the shallower events occurring

later, indicating a generally upward migration (Figs. 4c and d). Although the relative timing of the largest magnitude event in a cluster is often used as a proxy for defining aftershock sequences, here in the southern cluster, the magnitudes of the events are low and primarily similar. Furthermore, the seismicity distribution does not follow Omori's decay law since our clustering analysis would otherwise have detected it (Ajala & Kolawole, 2023).

## 5 Discussion

### 5.1 Crustal Softening in the Rukwa-Tanganyika Rift Zone

Brittle deformation in the crystalline crust, including fault- and folding-related damage, commonly creates zones of decreased bulk crustal density, manifested as zones of anomalously low Vs and high Vp/Vs ratio (Allam et al., 2014; Fang et al., 2019). Similarly, regions where brittle damage hosts melts/volatiles, and/or upwelling hydrothermal fluids are associated with relatively higher Vp/Vs values (e.g., Chatterjee et al., 1985; Nakajima et al., 2001; Hua et al., 2019). In active rift settings with absent surface volcanism, understanding the spatial distribution of upper-crustal seismic velocities permits the identification of mechanically weakened zones where tectonic strain may be preferentially localized. Delineating these near-surface structures will help to better understand how the crust accommodates tectonic strain along actively propagating rift basins and predict ground motion amplification during large earthquakes (e.g., Cormier and Spudich, 1984; Ajala and Persaud, 2021).

In the Rukwa-Tanganyika Rift Zone, two of the three areas of the highest upper-crustal Vp/Vs ratios (A1 and A2) occur at rift tips where syn-rift sedimentary cover is thinnest, and basement exposures dominate the surface geology (Figs. 1b and 3i). These anomalies occur at or near geothermal anomalies (hot springs and high heat flow sites in Figs. 1a, 3i) and are collocated with earthquake clusters and distributed normal faults. The anomalous seismicity cluster at the tip indicates the focus of active brittle deformation of the crystalline crust in a region that is lacking well-developed rift basins. At the Rukwa Rift tip and further to the northwest, the faulting pattern is generally characterized by distributed fault scarps that continue outboard from the border faults into the rift interaction zone (Fig. 1a). At the Mweru-Wantipa Rift tip, the rift faults appear to mainly cluster near the southeastern rift margin. Thus, we interpret the occurrence of the high upper-crustal Vp/Vs anomalies at the modern rift tips to represent a zone of mechanically weakened crystalline crust.

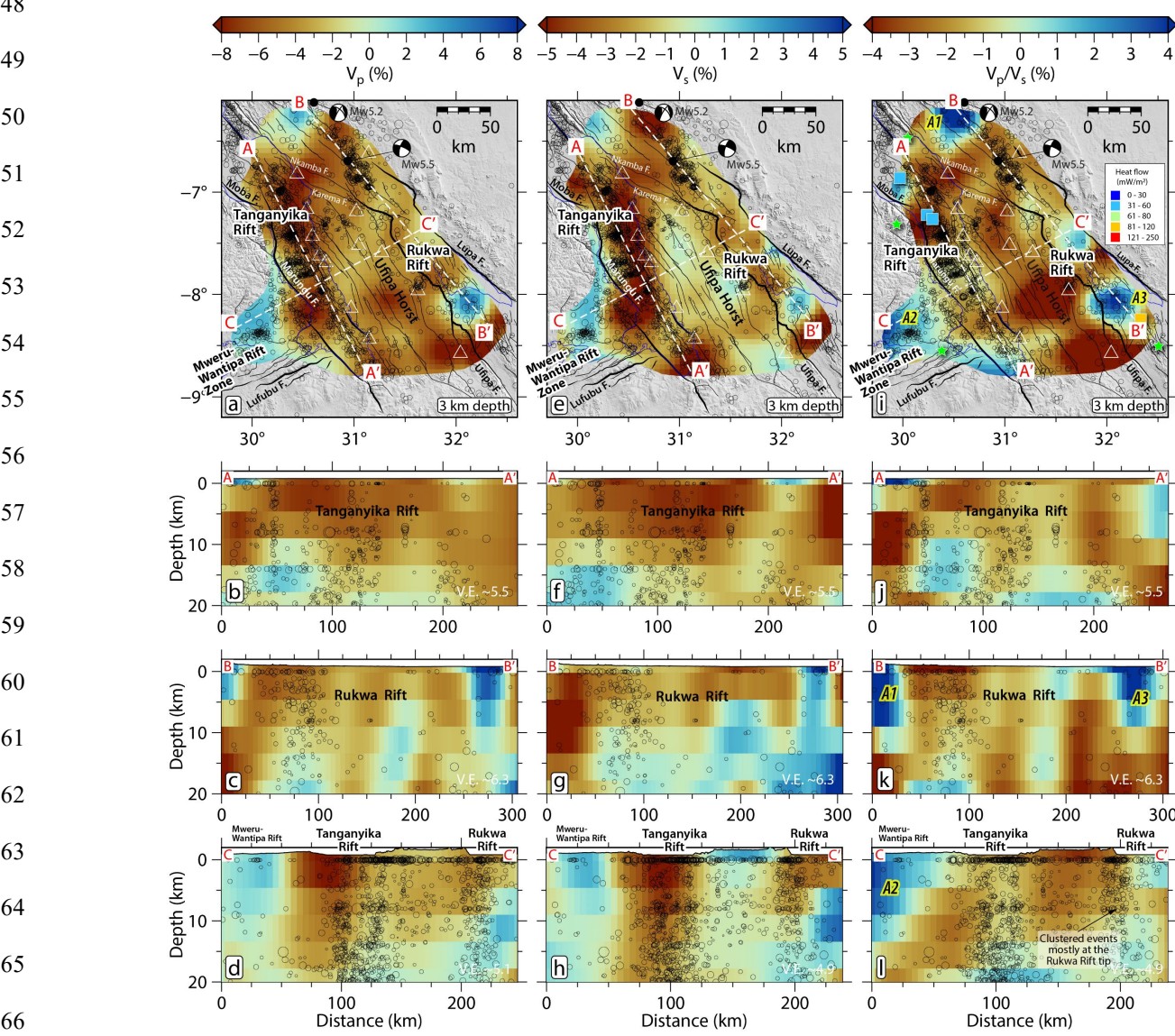

**Figure 3**. Maps and profiles through the tomographic models showing the perturbations relative to the starting models in Fig. 2b. (**a**) 3 km depth slice through the P wave velocity model. Unreliable areas of the model are not shown. Dashed black lines show the profile locations in b-d. (**b** – **d**) Profiles of the P wave velocity model. (**e** – **h**) Same as a-d but for the S wave velocity model. (**i** – **l**) Same as a-d but for the Vp/Vs ratios. Absolute values of the model parameters are shown in Fig. S16. Note that the geothermal center near anomaly A1 is north of latitude 6°S which is outside of the map coverage (see Fig. 1a).





## 5.2 Mechanical Weakening of Rift Tips: The Roles of Bending Strain and Crustal Fluids

The development of mechanically weakened crust at active rift tips reflects a critical rift process relevant for understanding how continental rifts propagate. This is analogous to microfracture propagation driven by high-stress concentrations at the crack tips (e.g., Kranz, 1979; Olson, 2004). Similarly, relatively large stress concentrations between interacting microcrack tips (Kranz, 1979) agrees with interpretation of stress concentrations within rift interaction zones that separate propagating rift tips (Kolawole et al., 2023). The northwestern tip of the Rukwa Rift is characterized by geomorphic features and tectonic deformation patterns that suggest an ongoing northwestward propagation towards the central and northern Tanganyika Rift (Kolawole et al., 2021a). The earthquake clusters at the Rukwa and Mweru-Wantipa rift tips (Fig. 2a) indicate that tectonic stresses and elastic strain concentrations are focusing on the rift tip zones. The brittle deformation field that is manifested by these earthquakes is likely accommodating the bending strain along the rift tip's flexural margin (Fig. 6a). Several studies have demonstrated that crustal bending due to accumulated fault displacement, glacial unloading, thermal subsidence, or sediment load induced crustal subsidence can focus significant strain in the upper crust, leading to brittle failure of the crust (e.g., Goetze and Evans, 1979; Stein et al., 1979; Nunn, 1985). Here, long-term accrual of fault displacement and sediment loading along the central hanging walls of the border faults causes basement down-flexure in the rift basin and proximal sections of the rift tip, and contemporaneous basement upwarping at the distal section of the rift tip (Fig. 6a). The crustal bending at the rift tips induces significant strain in the upper part of the brittle lithosphere, which may explain the prominent occurrence of earthquakes at the rift tips, best expressed at the northwestern tip zones of the Rukwa Rift (Figs. 2a, 2d). Since there is no data on the border fault displacements or basement depth variations from the rift axis into the areas of exposed basement ahead of the rift tip, we cannot provide a detailed analysis of how the changes in basement flexure imposes extensional vs contractional strain on the upper crust. Nevertheless, we suggest that damage clustering at a propagating rift tip is a relevant fundamental process that may facilitate mechanical weakening at the tips of active continental rifts.

In addition to bending strain-related earthquakes in the crust, the temporal and upward linear trends of low-magnitude seismicity migration in the upper mantle beneath the proximal rift tip in the Rukwa Rift tip (Figs. 4b-f) suggest fluid (volatiles) related earthquake triggering. We interpret that the northern cluster likely represents fluid-induced microseismic creep due to the fast migration velocity (>1 km/hr) (Fig. 4e), and the southern cluster likely indicates fluid flow due to the much slower linear migration velocity (Fig. 4f) (e.g., Zhang & Shearer, 2016). These results are further corroborated by the high waveform similarity of the events recorded at nearby stations (Figs. 4g and h; Raggiunti et al., 2023). In the Tanganyika Rift, the detected spatiotemporally clustered events extend up from the moho to the upper-crust (Fig. S24), and the events are primarily in the crust beneath the Mweru-Wantipa Rift tip (Fig. S24e). Although primarily hosted in the crust, we interpret that the detected clustering events in the Tanganyika and Mweru-Wantipa are likely also triggered by fluids, and that the fluids are potentially related to both mantle and hydrothermal sources. Thus, our cluster analysis results are consistent with previous studies that suggest the presence of partial melt in the crust beneath Tanganyika Rift Zone (Hodgson et al., 2017; Lavaysseier et al., Ajala et al., 2024).

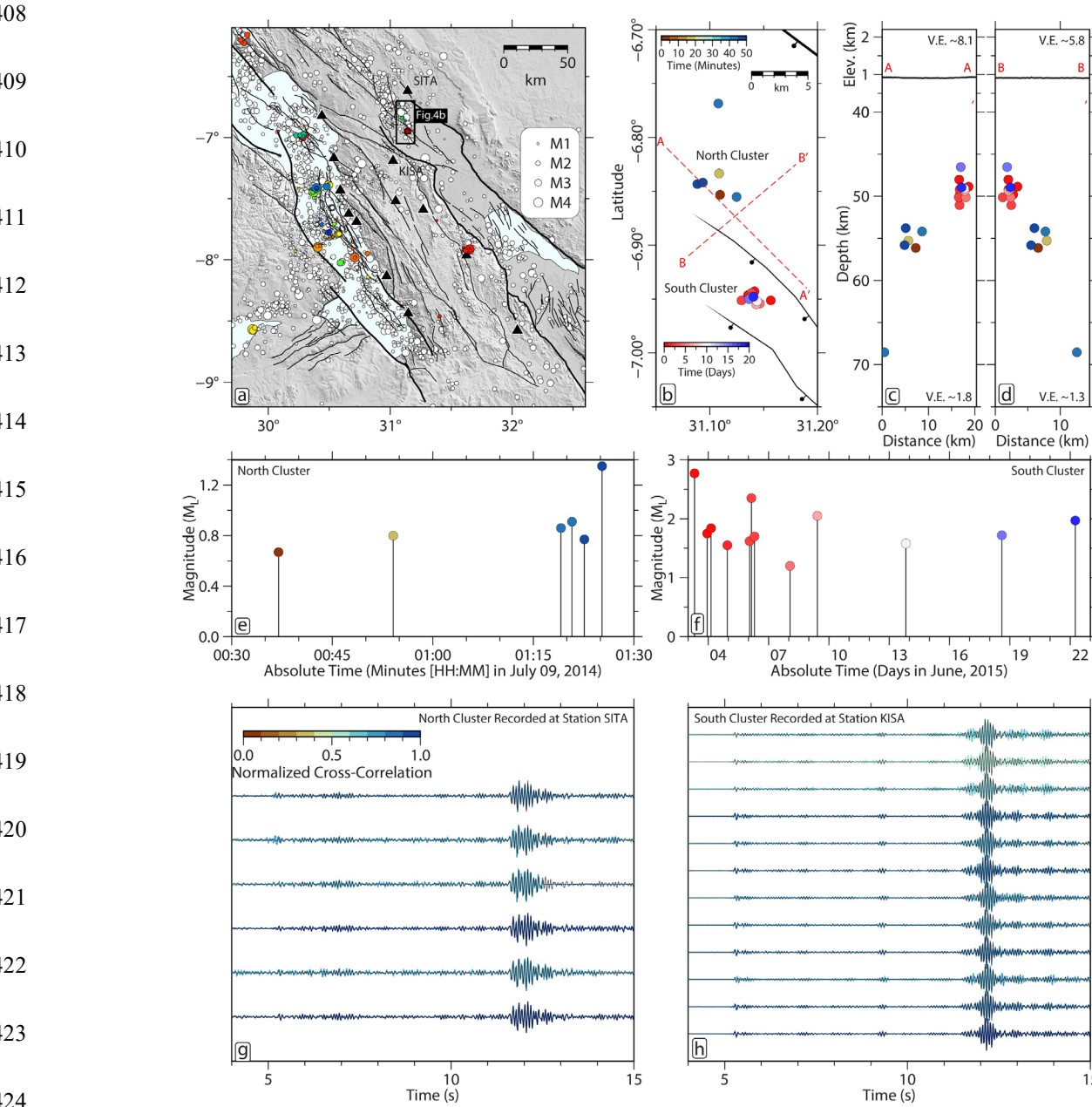

**Figure 4**. Delineation of spatiotemporal seismicity clusters with a focus on the Rukwa rift tip swarms. (**a**) Map of the study area showing the broad seismicity and detected spatiotemporal clusters (colored circles). Red polygon indicates the frame for the clusters shown in panel b. (**b**) Zoom-in map of the two clusters at the Rukwa tip color-coded according to their occurrence in time relative to the parent event (i.e., first event) in each group. Red dashed lines are the locations of the cross-sectional profiles in panels c and d. (**c**) Rift-parallel, and (**d**) Rift-perpendicular profiles showing the projected clusters. (**e**) Magnitude-time plot for the northern cluster events. (**f**) Magnitude-time plot for the southern cluster events. (**g**) 2–15 Hz waveform records for the north cluster events recorded at station SITA highlighted in panel a. Each waveform is colored using the normalized cross-correlation coefficient computed by comparing the similarity of each waveform in the sequence to the waveform of the parent event. The parent event waveform is also plotted on all the waveforms as a black line for visual comparison. All traces have been time-shifted to maximize the correlation. The maximum cross-correlation value occurs for the first trace

since it represents the autocorrelation (correlation of the parent waveform with itself). (**h**) Similar to panel g but for the southern cluster
recorded at station KISA.

The spatial relationship between large earthquake clustering and the velocity distribution in the upper crust may provide insight
into how bulk rock alterations may influence seismicity and strain accommodation at the rift tips. The occurrence of geothermal
anomalies in the vicinity of the high Vp/Vs ratio anomalies suggest that ascending fluids may be advecting heat into the upper
crust. At both the Rukwa and Mweru-Wantipa Rift tips, we observe that the most prominent seismicity clustering occurs near
the margins of the high Vp/Vs ratio upper-crustal anomalies, and not within the anomalies (Figs. 3i,k,l). We infer that this
pattern indicates frictionally stable conditions promoting aseismic failure within the crustal blocks of high Vp/Vs ratio, and
frictionally unstable conditions promoting seismic failure in their surrounding crust. The brittle failure of brittle discontinuities
may be aseismic or seismic depending on confining stress, temperature, and compositional characteristics of the crust and the
fault rocks they host (e.g., Blanpied et al., 1991; Carpenter et al., 2011; Kolawole et al., 2019). Given the same loading
conditions around the rift tips, it is possible that significant fluid-rock alterations of the crust due to the migrating fluids within
the areas of highest Vp/Vs resulted in frictionally stable conditions within the zones of highest Vp/Vs ratios (D1 in Fig. 6a) as
opposed to their surrounding regions that are failing by seismogenic deformation (zone D2). Within the central regions of the
Rukwa Rift, the Vp/Vs anomaly A3 is collocated with an area of relatively less intra-rift fault occurrence (Fig. 5i) but is in the
hanging wall of the Ufipa border fault near a known geothermal anomaly (Jones, 2020). Since A3 is confined to <5 km depth
(Fig. 3k), it may also represent a compositionally altered and mechanically weakened section of the border fault and its hanging
wall block, similar to velocity anomalies observed near geothermal field of active rifts elsewhere (e.g., Hauksson and Unruh,

455 2007).


Although our results generally indicate active deformation at the propagating rift tips of the Rukwa and Mweru-Wantipa rifts,
the relatively greater abundance of data at the Rukwa Rift tip permit a characterization of how the controls on the deformation
may vary from the proximal rift tip zones to the distal tip zones. The depth distribution of seismicity and the detected
spatiotemporal seismicity clusters, the along-rift variation of crustal thickness, and relative location of high Vp/Vs anomaly
suggest that the proximal tip zones (tip zone 1) is dominated by upper-crustal, lower-crustal, and upper-mantle deformation
(Fig. 5). However, the crust appears to thicken towards the distal tip zones (tip zones 2 to 3) and the seismicity patterns appear
to become shallower and primarily focusing on the upper crust at tip zone 3 (Fig. 5). In general, we infer a through-going
crustal deformation in the proximal rift tip zones controlled by crustal thinning and infiltration of volatiles into the crust with
focused crustal bending strain (synclinal?), all of which transition into a dominantly upper-crustal deformation at the distal tip
zones where bending strain (anticlinal) and fluid-rock alterations control the brittle deformation. Published models for rift
linkage demonstrate that rift basins can propagate laterally and interact when in proximity (e.g., Allken et al., 2012; Corti,
2012; Molnar et al., 2019; Nelson et al., 1992; Zwaan et al., 2016; Zwaan and Schreurs, 2020; Neuharth et al., 2021; Kolawole
et al., 2021a). Models also show that laterally propagating rift tips may host stress concentration zones (van Wijk and
Blackman, 2005; Le Pourhiet et al., 2018). Our study presents evidence from a natural rift for the first time, revealing the
presence of crustal weakening at a laterally propagating continental rift tip, and in addition, shows how the weakening is likely
controlled by a combination of crustal bending strain and fluids (ascending volatiles and migrating hydrothermal fluids). We
propose a model for lateral rift propagation whereby progressive rift tip propagation is marked by the development of localized
weakened crustal at the rift tip (Time T1, Fig. 6a) which subsequently gives way to a lengthened rift basin (Time T2, Fig. 6b).

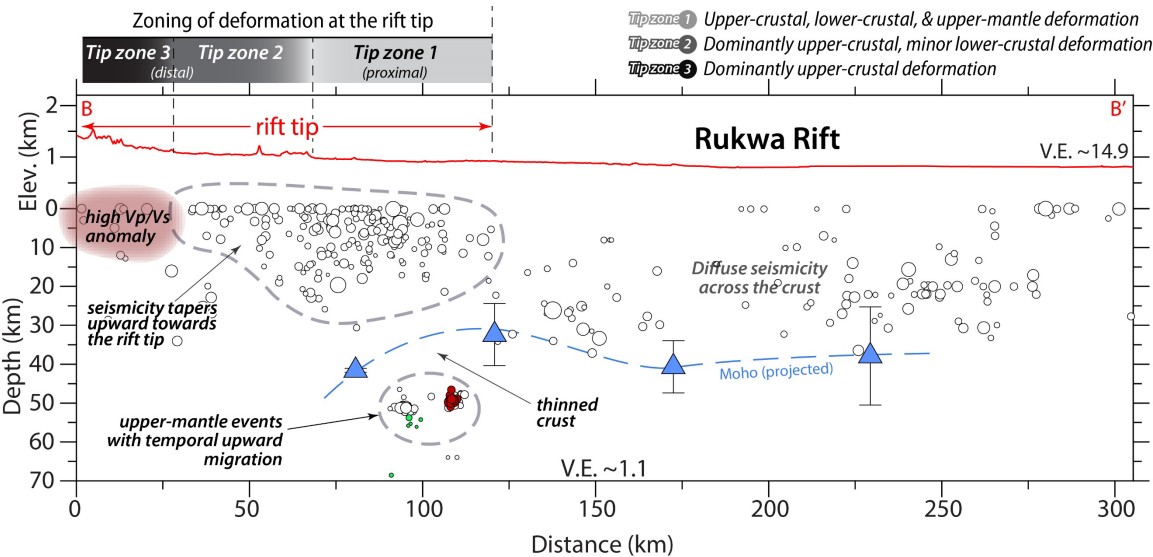

**Figure 5**. Interpretation of longitudinal cross-sectional profile B-B' of the Rukwa Rift (same as in Fig. 2d and S24d) highlighting the spatial relationships between the broad seismicity distribution, detected fluid-related clustered events (colored upper-mantle events), upper-crustal low-velocity anomalies, moho depth distribution, and the zoning of active deformation at the rift-tip.


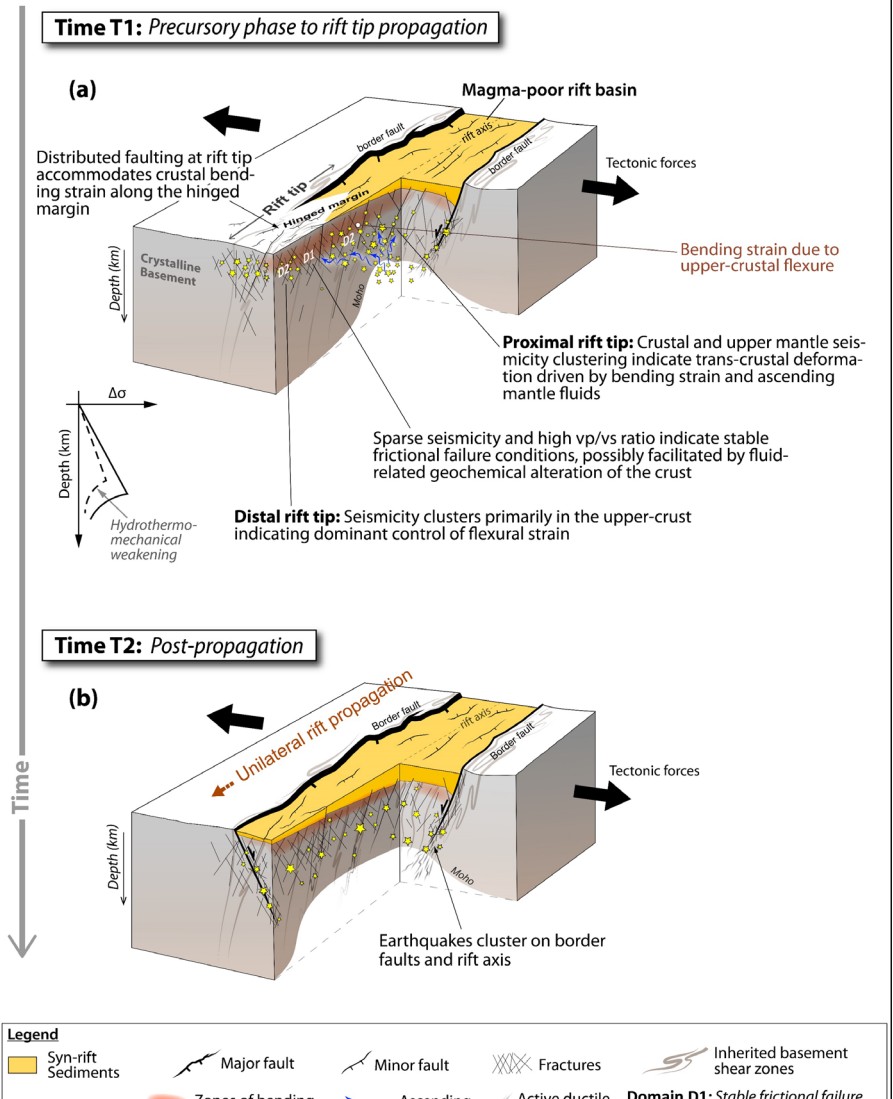

**Figure 6**. Cartoons showing the proposed model of crustal strain accommodation during the unilateral propagation phases of active continental rift tips, based on the results of our study. Note that the panel b of the cartoon is idealized to speculate a likelihood of decreased seismicity at a paleo-rift tip zone post propagation of the rift tip and does not include earthquakes occurrence due to other tectonic processes that may promote strain localization within the rift axis.

## Conclusions

To understand how tectonic strain is accommodated along actively propagating magma-poor continental rifts, we constructed three-dimensional velocity models of the crystalline crust beneath the Rukwa-Tanganyika Rift Zone where the Tanganyika Rift is interacting with the Rukwa and Mweru-Wantipa rifts. The results show anomalously high Vp/Vs ratio anomalies at the Rukwa and Mweru-Wantipa rift tips and their rift interaction zones with the Tanganyika Rift, representing, for the first time, geophysical evidence demonstrating crustal softening of rift tips in a region of active unilateral rift propagation. We detect distinct earthquake families within the deeper rift-tip seismicity clusters that exhibit linear upward migration patterns, and temporal evolution patterns that suggest fluid migration and associated creep failure. We determine that brittle damage due to bending strain and thermomechanical alteration of the crust by ascending fluids (mantle-sourced volatiles and hydrothermal fluids) are accommodating the mechanical weakening at the rift tip to facilitate the propagation of the rift tip into unrifted crust within the rift interaction zones. Furthermore, we observe a transition from collocated thinned crust and through-going crustal and upper-mantle seismicity in the proximal tip zones, to dominantly upper-crustal seismicity in the distal tip zones, indicating an along-axis variation in the controls on rift tip deformation. The results of this study provide new and compelling insights into how continental rift tips propagate, link, and coalesce to form continuous axial rift floors — a necessary ingredient for initiating large-scale continental break-up axes.

## Acknowledgments

This project was supported by funds from the Columbia Climate School and the Vetlesen Foundation awarded to Folarin Kolawole. We thank Finnigan Illsley-Kemp, Frank Zwaan, and anonymous reviewers for comments that helped to improve the earlier and later versions of the manuscript. Some figures are plotted using GMT (Wessel et al., 2019).

## Author contributions

F.K. and R.A. conceptualized the project. R.A. performed the modeling. F.K. and R.A. interpreted the results. F.K. wrote the manuscript. R.A. revised the manuscript.

## Competing interests

The authors declare no competing interests.

## Open Research

Computer programs and files to reproduce our results are in Ajala and Kolawole (2023).

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
