# Peer review of "Crustal Softening at Propagating Rift Tips, East Africa"

_EGUsphere, 2023_

## Author Response (AR1)

**We thank topic editor Frank Zwaan for handling our paper, and the two anonymous reviewers for their useful comments have helped to improve our manuscript. We have addressed all the comments of both reviewers and the topic editor and implemented the recommended revisions where necessary. Below, we provide a line-by-line response (in red) to the review comments (in gray). Kindly note that the line numbers referenced in our replies are from the revised version of the manuscript.**

**Reviewer #1**
In this paper, the authors image the velocity structure of the Rukwa-Tanganyika rift zone to investigate the processes responsible for rifting propagation at the rifts' tips.Overall, this paper is well written and will contribute to a better understanding on rifting propagation in the crust. I just have a few comments on the model testing, and I feel the discussion section could be further developed, but I very much enjoyed reading this manuscript.

We thank the reviewer for their comments and suggestions. We have implemented the suggested in-text changes and we have replied to the suggested changes in the old manuscript (also attached). We respond to their specific comments below.

*Specific comments:*
Although I appreciate seeing all the checkerboard tests, generally I have the impression that you tested the anomalies you wanted in your synthetic models. For me,the models in Fig 3 do not really correspond to the results in Fig 5. For example, in Fig 5 in the sections b, f and j there are anomalies that you do not use in your tests. Is it because the high Vp/Vs are not high enough? What about the strong low anomalies? Shouldn't they be included in the tests too? Does it change the final results if you add the low anomalies or the smaller anomalies?

Yes, we only tested the anomalies of interest. This is standard practice in seismological structural imaging. Simply put, the conventional goal of synthetic tests is to test if the features of interest interpreted from seismic models are reliably recovered by the illumination afforded by the seismic array. In this specific case, we are primarily interested in testing the recoverability of the high Vp/Vs anomalies at the rift tips.

In Fig 5, sections c and g, the high Vp and Vs in the east seem to be stronger close to the surface and at 20 km depth, can you modify your tested anomaly to reflect this? Does it change anything?

We could modify the intensity of the synthetic anomalies at different depths. The checkerboard models test exactly this component. It would be another test, but we do not think it would significantly modify the results.

In your synthetic model Fig 3, the Vs anomalies are both lower Vs, but in your preferred inversion, there is a low anomaly in the north and a high in the south. Why not reflect this in your synthetics?

As mentioned earlier, the goal of a synthetic test is not to reproduce every detail in the real inverted model but only the most important features.

As you use your preferred model results to do the synthetic models, I would present the inversion first. Maybe switch sections 3.3 and 3.4?

We could, but it seems a better flow would be presenting the methods before the results. Unfortunately, synthetic modeling falls under the methodology.

It is lacking a bit more explanation on why it is your preferred inversion.

This is somewhat subjective as the inversion results from the three different starting models show similar features.

You discuss the high Vp/Vs anomalies but not the low, can you explain why? Can they not bring information too?

We agree that low Vp/Vs anomalies could convey useful information. However, in the context of the current geologic problem explored in our study, and to avoid unnecessary sidetracking, we decided to focus on the higher-than-average Vp/Vs anomalies. This is also quite common in seismology as high Vp/Vs anomalies highlight the presence of crustal fluids, thick sediments etc.

The discussion section could be further developed. In my opinion there are two missing discussions. Can you compare with other rifts in the world? And you do not mention fluids. However, there was evidence in Lavayssière et al. (2019) that the deep earthquake clusters at the northern tip of Rukwa were possibly due to fluids. Can you comment on this with your new results? Does it bring another hypothesis for rift propagation?

A key component of our interpretations of the model results is that the focused crustal deformation at the Rukwa and Mweru-Wantipa rift tips is influenced by bending strain and thermomechanical fluid action. However, we agree that the presence of upper-mantle earthquakes near the rift tip indicates that the fluid action may not necessarily be hydrothermal, but may have a significant component of mantle-sourced fluids (i.e. melt) as the reviewer has mentioned. We have another study underway that is focused on spatiotemporal cluster analysis of the earthquakes as the complete analysis of the earthquake catalog for tight spatiotemporal clustering and its mechanisms is beyond the scope of the current manuscript. However, we do agree with the reviewer that it would improve the strength of the current manuscript if we include some of the cluster analysis. Therefore, we have now incorporated a simple clustering analysis of the rift tip seismicity clusters which support the interpretation of an influence of fluid action on active deformation of the rift tip. We have also expanded the results and discussion sections of the manuscript to include the explanation of the cluster analysis and implications for fluid migration activity at the rift tips.

On the comparison between rifts... this is beyond the scope of this study as it would require conducting a similar study in other active rift basins. To the best of our knowledge, the analysis of rift tip deformation presented in the manuscript is the first of its kind. We anticipate that our paper will inspire geophysical investigation of rift tip processes in other active rift systems.

**Technical corrections:**

Previous comments and smaller typing and figure comments have been added directly in the manuscript pdf version and on the supplementary pdf.

We thank the reviewer for the effort. We have updated the manuscripts with the suggested corrections where necessary.

**Reviewer #2**

Understanding the connection between two propagating rifts is crucial to fully capture the formation of a continuous, well developed plate boundary. To this end, Kolawole and Ajala use three-dimensional velocity model and distribution of seismicity and active faults to capture the propagation of the Tanganyika and Rukwa rifts in the western rift of the East Africa Rift system. The authors applied appropriate methods to answer the research questions. The MS is well written, and the figures are of high quality. While I find this study quite interesting, I think a rigorous analysis on the impact of fluid migration in assisting the linkage enhances the impact of their study.

We thank the reviewer for their comments and suggestions. We agree with the reviewer's suggestion of adding an analysis of the earthquakes for insights into fluid migration patterns. Within the limits of uncertainty of the recorded seismic events, we have provided the relevant spatiotemporal clustering analysis that thus support the presence of ascending fluids from the mantle up into the crust at the Rukwa Rift tip.

We respond to their general comments below.

*General comments*

The Tanganyika and Rukwa are two continental rifts situated in a region where the lithosphere has a thickness of about 150 km. In such tectonic settings, numerical models suggest that rifts attempt to link, instead of propagating along strike in their separate ways (e.g., Neuharth et al., 2022). This observation is supported by the present authors suggesting that the Rukwa rift is propagating to the NW to link with the central and northern Tanganyika rift by breaking the topographically elevated region between them. The question is whether this propagation is facilitated by purely tectonics forces or not (i.e., assisted by fluid migration). After studying the distribution of seismicity, Lavayssière et al., 2019 argued that rift bounding faults penetrated the entire crust, which provide an easy access for the fluid from the mantle to migrate to the shallower depth. This hints that the propagation of the rifts in the region can be assisted by fluid migration in the crust. This hypothesis has to be tested by looking at the swarm activity (i.e., fluid induced earthquakes are characterized by very similar waveforms) of the earthquakes before concluding that the rift propagation is purely tectonic and more complex mechanisms (line 402-411) are proposed. We should keep in mind that fluid saturated geological units are characterized by an increase in Vp/Vs (e.g., line 356-358).

We agree. A key component of our interpretations of the model results is that the focused crustal deformation at the Rukwa and Mweru-Wantipa rift tips is influenced by bending strain and thermomechanical fluid action. However, we agree that the presence of upper-mantle earthquakes near the rift tip indicates that the fluid action may not necessarily be hydrothermal, but may have a significant component of mantle-sourced fluids (i.e. melt) as the reviewer has mentioned. We have another study underway that is focused on spatiotemporal cluster analysis of the earthquakes as the complete analysis of the earthquake catalog for tight spatiotemporal clustering and its mechanisms is beyond the scope of the current manuscript. However, we do agree with the reviewer that it would improve the strength of the current manuscript if we include some of the cluster analysis. Therefore, we have now incorporated a simple clustering analysis of the rift tip seismicity clusters which support the interpretation of an influence of fluid action on active deformation of the rift tip. We have also expanded the results and discussion sections of the manuscript to include the explanation of the cluster analysis and implications for fluid migration activity at the rift tips.

In your abstract, you stated that the tips are characterized by little sedimentation. So, how can we expect the sediment load to cause crustal down-flexure at the tips (line 406-407)?
The referenced lines state that "long-term accrual of fault displacement and sediment loading on the hanging walls of the Lupa and Ufipa border faults causes crustal **down-flexure in the rift axis and contemporaneous crustal upwarping at the rift tips**".

Line 23 – I would use "crustal block" instead of "basement block"
Agreed. Done

Line 33 – The MS is written in such a way that that it studies the linkage between Tanganyika - Rukwa rifts, which appears to contradict with what you have mentioned in the abstract (i.e., tips of Rukwa, Mweru-Wantipa, and Tanganyika rift).
It is true that we focused mostly on the Rukwa-Tanganyika rift interaction zone than on the Tanganyika-Mweru-Wantipa rift interaction zone. We consider both to be actively deforming rift interaction zones within the Rukwa-Tanganyika Rift Zone. However, the greater focus on the Rukwa-Tanganyika rift interaction zone is because the area has significantly more data, more previous studies, and is better understood than the Tanganyika-Mweru-Wantipa interaction zone. In addition, we have better seismic station coverage on the Rukwa/Tanganyika side of the rift zone and no stations on the Mweru-Wantipa side of the rift zone.

Line 135 – should be mW/m
The unit of heat flow ('heat-flow density') is $mW/m^2$
i.e. the flow of energy per unit area per unit time

You suggested that the propagation at the tips is accompanied by distributed faulting (e.g., line 389), however in line 474 & 476, it reads that localized crustal weakening occurs at the tips of the rifts.
We have reworded the text to "...revealing the presence of crustal weakening at a laterally propagating continental rift tip".

Line 470-474 – the discussion on rift stalling does not go well with your rift propagation study. It is confusing.
We agree. We have now removed the text.

I am wondering how the ~NE oriented Mweru-Wantipa rift opens under the ~N80°E regional stress field? Is there a rotation of the stress field around this rift?
This is a good question and it is our opinion that the faults are not favorable for reactivation in a 080° regional SHmin. The Mweru Rift, which is the continuation of the Mweru-Wantipa Rift to the southwest, is shown to be have a 118° regional SHmin (Delavaux & Barth, 2010) which may be a more favorable stress field for reactivation of the Mweru-Wantipa faults. We have included this information in the introduction section of our manuscript.

**Topic Editor's Comments**
Two reviewers have provided comments on the manuscript, and I would like to invite the authors to reply to each of these comments (note that reviewer 1 has also uploaded an annotated PDF with comments). Next to that, I have an additional important comment for the authors to consider as well. A big issue I see is that there seems to be no clear methods section. The text kind of transitions from introduction and description of the study area into results, without it being clear where the results really start. The authors should introduce a methods section after the introduction that clearly states what methods and techniques were used, before presenting the new results. I believe this can be easily solved though.
Done. We have now revised the manuscript to include a dedicated methods section that is separate from the results section.

Other minor points (come may overlap with those of the reviewers):
Line 61: do the authors mean "none of the sub-basins" or "not all of the sub-basins"?
The latter. Sentence reworded to "…not all the basins record the three phases of…"

Line 62: wording: perhaps use "that contains" instead of the somewhat complex "where there exists"
Sentence reworded to "…Within the rift zone, the Rukwa Rift is the only basin with basement-penetrating borehole logs to constrain…"

Line 109-100: the Daly et al. reference seems to be missing in the list
Reference added

Line 110: is this the local extension direction, or the regional plate motion/divergence direction? These are not necessarily the same directions.
Regional extension direction. Local extension direction is associated with local strain field on each fault (inferred from geological strain measurement or earthquake focal mechanism for a single slip surface or fault segment), whereas, regional extension direction is at the scale of a rift zone and is inferred either from the inversion of many geological strain measurements or earthquake focal mechanisms across the entire rift zone. The regional extension direction may also be referred

to as the plate motion/divergence direction as measured from e.g. GNSS-based geodetic plate motion vectors.

Perhaps mention also the rotation of the Victoria plate, which may cause the difference in extension orientation between north and south?

We would prefer not to add this as the control on the variation in extension direction is not well understood. For example, strong crustal mechanical anisotropy has been proposed for Rukwa/Tanganyika Region (see Morley, 2010). The anisotropy control seem to be compatible with the control of inherited basement fabrics on the trends of the northern Tanganyika and southern Tanganyika/Rukwa rifts (NW-trending Ubendian Belt in the south and N-to-NNE trending Karagwe-Ankolean Belt in the north; e.g., Muirhead et al., 2019; Kolawole et al., 2021; Shaban et al., 2023).

Line 114: There is only one large fault that seems to be changing polarity on the map it seems? (at the Kavala Island ridge? This may need some rephrasing.

We beg to differ. The rift is segmented, defined by distinct border faults that alternate polarity. This is well documented in literature (e.g., Versfelt and Rosendahl, 1989; Muirhead et al., 2019; Shaban et al., 2023).

Line 117: in contrast to the Tanganyika and Rukwa Rifts, the Ufipa Horst itself is not indicated in a figure I believe? It may be good to do so for clarity.

It was in Fig 5 panels. We have now added the label to Figs. 1 and 2.

Line 120: it is roughly westward but not quite westward, perhaps say WNW-ward instead?

Done

Line 121: perhaps start this sentence with "to the SW" or so, because the first time I read it, I thought we were still talking about the Ufipa Horst area, and got lost for a moment while looking for the Mweru-Wantipa rift there.

Done

Line 123: that should be Fig. 2a I believe?

No, Fig. 1a (Lufuba Fault was labeled).

Line 174: is it "rift overlap zone" or "rift interaction zone" (terminology to be adapted throughout the text)? Perhaps this terminology should already be introduced at the start of the introduction, where this kind of structure is mentioned?

We agree. Corrected

Line 176: it would be good to clearly specify that this elevated region is the Ufipa Horst

The elevated region is much broader than the Ufipa Horst. The host is the southern portion of the rift interaction zone. The other region is north of the Ufipa Horst, encompassing the region between the northern tip of the Rukwa and the eastern flank of the central Tanganyika Rift. For better clarification, we have now added this explanation to the text.

Line 178: is this Kolawole et al (2021a) or (2021b)?
Corrected (2021a).
Line 183: "rift tip" is somewhat poorly visible (as are the A, A' etc. in the sections). See also comments on color use in Fig. 2 below
We have enlarged the texts

Line 185: here, the text kind of jumps again to the Mweru-Wantipa rift situation. See comment on line 121 on how a smoother transition could be made
Transition text added

Line 334: figure reference order seems incorrect?
Corrected

Line 337: the authors may mean Fig. 5 here?
Yes. Corrected

Line 339: slower or lower VP?
Corrected to 'lower'

Line 392: "outboard" is maybe not the best word? The faults seem to fall in the extension of the Rukwa zone, this I would say "inboard"?
We retain 'outboard', and for clarity, we have now reworded the text to "distributed faults that continue outboard from the border faults into the rift interaction zone".

Line 402-411: if there are lots of quakes due to bending of the crust at the tip of the rift, should there not also be a zone closer to the rift, where the crust is being bent again (to horizontal level)?
Hypothetically, yes. Since there is no data on the border fault displacements or basement depth variations from the rift axis into the areas of exposed basement ahead of the rift tip, we cannot provide a detailed analysis of how the changes in basement flexure imposes extensional vs contractional strain on the upper crust. While this is a next step in understanding strain accommodation at propagating rift tip, it is beyond the scope of this paper.

Line 483-484: isn't this initial crustal softening in itself not already a form of rift propagation (it already starts deforming). Perhaps the term "rift propagation" needs to be clearly defined in this context.
We agree. We have reworded the texts across the manuscript to state that the softening itself represents the onset of propagation. We agree that the propagation of the Rukwa Rift tip is underway already, however, since the softened zone cannot be dated and the surface is still basement dominated, we infer that the softening happens early in the process.

Line 486: it should be "axes" I believe?
Corrected to 'axes'.

■ Fig. 1.

Note that some of the annotation is poorly visible (to me, I have slight red-green color blindness) the orange on pinkish background used for text and terrane boundaries in panel (b). The same for the pinkish earthquake dots, arrows and stars in panel (a).
We thank the reviewer for bringing this issue to our attention. We take such issues seriously and we have updated our figure as recommended.

Also, please make sure all abbreviations used in the figures are specified in the caption (e.g. Ub, Wa, Ka etc. in panel b. NB: I now see that these are provided in the legend, but it may be better to put them in the caption, where other abbreviations are specified as well □ they seemed to be missing there).
Done.

Note that the scale in panel (a) is at a different place than in panel (b), and also the north arrow is at a different place, I would suggest using the same location for both in both panels.
Done

■ Fig. 2.
Also here, some of the annotation is poorly visible (I have trouble distinguishing the traces of the profiles and the letters indicating them, as they seem to be very faint).
We have fixed this issue.

■ Figs. 3-4:
Note that in the sections, part of the annotation is overlapping with the line indicating the surface and is thus poorly visible
We have fixed this issue

Panels (a), (e), (i), perhaps use "3 km depth" to avoid any confusion it could be a horizontal scale
Done

■ Fig. 5:
This figure has a lot of detail, but is somewhat small. Could the individual panels be made a bit larger? Perhaps it would be possible to split the figure in two or three parts to allow each individual part of the larger?
Done

Note that (to me) the traces of the profiles are poorly visible (red on a reddish background), same for the pink stars on a greyish background
We have changed the traces of the profiles to white dashed lines and they show up better now. The colors of the stars have also been changed.

In section view, some of the annotation is not well visible
We have now fixed this

---

## Author Response (AR2)

**We thank topic editor Frank Zwaan for the additional comments on our manuscript. We think they are useful and have helped to improve the paper. We have addressed all the comments and implemented the recommended revisions. Below, we provide a line-by-line response (in red) to the review comments (in gray).**

Additional Changes

In addition to the figure edits suggested by the topic editor, we made a slight rewording of the title in order to better capture the essence of the major contributions of our paper. The new title is 'Propagating Rifts: The Roles of Crustal Damage and Ascending Mantle Fluids'.

Topic editor comments

Red dotted lines and red text in Fig. 1b (can all be black, I would say).

Done.

Red dots in Fig. 2 (maybe use circles filled with light grey for all the quakes, and black instead of red for the dots)

Done

Red outline and, to a degree, the red dots in Fig. 4a (outline can be made black, I have no ready solution for the red dots, so perhaps it is ok to keep it this way)

Done

Red stars in Fig. 6 (perhaps make them yellow with black outline or so?)

Done